# STABLE ANISOTROPIC REGULARIZATION

**William Rudman**
Department of Computer Science
Brown University
william_rudman@brown.edu

**Carsten Eickhoff**
School of Medicine
University of Tübingen
carsten.eickhoff@uni-tuebingen.de

## ABSTRACT

Given the success of Large Language Models (LLMs), there has been considerable interest in studying the properties of model activations. The literature overwhelmingly agrees that LLM representations are dominated by a few "outlier dimensions" with exceedingly high variance and magnitude. Several studies in Natural Language Processing (NLP) have sought to mitigate the impact of such outlier dimensions and force LLMs to be isotropic (i.e., have uniform variance across all dimensions in embedding space). Isotropy is thought to be a desirable property for LLMs that improves model performance and more closely aligns textual representations with human intuition. However, many claims regarding isotropy in NLP have been based on the average cosine similarity of embeddings, which has recently been shown to be a flawed measure of isotropy. In this paper, we propose I-STAR: IsoScore$^\star$-based STable Anisotropic Regularization, a novel regularization method that can increase or decrease levels of isotropy in embedding space during training. I-STAR uses IsoScore$^\star$, the first accurate measure of isotropy that is both differentiable and stable on mini-batch computations. In contrast to several previous works, we find that *decreasing* isotropy in contextualized embeddings improves performance on most tasks and models considered in this paper. [1]

## 1    INTRODUCTION

Several previous works have investigated the role of isotropy in Large Language Model (LLM) representations (Rudman et al., 2022). A distribution is *isotropic* if the variance of the data is uniform and the data dimensions are uncorrelated. In practice, a distribution is isotropic when its covariance matrix is proportional to the identity matrix. Studies have found that representations from LLMs, such as BERT or GPT-2, lack the property of isotropy and that contextualized word embeddings are dominated by a few "rogue" or "outlier" dimensions (Timkey and van Schijndel, 2021; Kovaleva et al., 2021). Several previous works have argued that *anisotropy*, i.e., the lack of isotropy, is detrimental to LLM embeddings as it 1) forces representations to occupy a "narrow cone" in space (Ethayarajh, 2019; Cai et al., 2021); 2) obscures linguistic information, thereby limiting the expressive power of the embeddings (Gao et al., 2019; Zhang et al., 2020; Mickus et al., 2019), and; 3) hinders performance on a variety of downstream tasks (Kovaleva et al., 2021; Biś et al., 2021; Timkey and van Schijndel, 2021). However, some recent works have challenged previously held conceptions about isotropy, arguing that current methods of measuring isotropy are fundamentally flawed (Rudman et al., 2022; Rajaee and Pilehvar, 2021a). To address these concerns, Rudman et al. (2022) propose IsoScore, an accurate and robust method for measuring isotropy based on the covariance matrix of a distribution. Although IsoScore is an effective method for measuring isotropy, we demonstrate that IsoScore is neither differentiable nor stable when the number of points in a given sample is small. Therefore, IsoScore cannot serve as an effective model regularizer.

Given the recent criticism of methods for measuring isotropy, a reassessment of previously accepted theories of isotropy in LLMs is needed. This paper aims to determine the relationship between isotropy and model performance on various language models and fine-tuning tasks. We first propose IsoScore$^\star$, a method for measuring isotropy that is 1) fully differentiable, 2) incorporates classical techniques for covariance estimation to create stable isotropy estimates for mini-batch data, and 3) approximates IsoScore for large sample sizes. We then use IsoScore$^\star$ to develop I-STAR: IsoScore$^\star$-based STable

---

[1]Code: *https://github.com/bcbi-edu/p_eickhoff_isoscore.git*

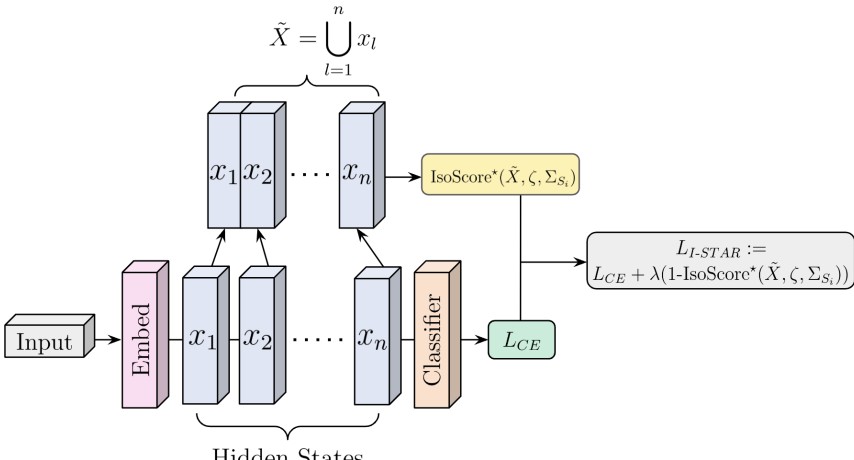

Figure 1: Forward pass of our I-STAR loss function. Let $x_l$ be the token embeddings in a mini-batch at layer $l \in \{1, 2, ..., n\}$, let $\tilde{X} = \bigcup_{l=1}^{n} x_l$, let $\Sigma_{S_i}$ be the shrinkage covariance matrix for epoch $i$ and let $\zeta \in (0, 1)$ be the shrinkage parameter. I-STAR loss is a weighted sum between cross-entropy loss, $L_{CE}$, and IsoScore$^\star(\tilde{X}, \zeta, \Sigma_{S_i})$ where $\lambda$ is the tuning-parameter. Negative values of $\lambda$ correspond to decreasing isotropy in representations, and positive values of $\lambda$ encourage isotropy.

Anisotropic Regularization. I-STAR is a flexible way to adjust isotropy in model representations during training or fine-tuning. In contrast to works that use "flawed" measures of isotropy, we find that using I-STAR to decrease isotropy in embedding space, i.e., making representations more *anisotropic*, tends to improve downstream performance across three different LLMs and nine different fine-tuning tasks. Our finding that anisotropic representations perform better on downstream tasks is aligned with literature outside of NLP that argues anisotropy is a natural by-product of stochastic gradient descent, where anisotropy helps models escape local minima in the loss landscape and, thus, generalize better to unseen data (Zhu et al., 2018). Additionally, our findings are supported by a well-established body of literature arguing that lower intrinsic dimensionality of network representations in later model layers corresponds to better performance on downstream tasks (Ansuini et al., 2019; Recanatesi et al., 2019; Chung et al., 2018). This paper makes the following novel contributions.

1. We propose IsoScore$^\star$, a robust method for measuring isotropy that is stable even when the number of samples in a point cloud is small.
2. We present a novel regularization method, I-STAR: IsoScore$^\star$-based, STable Anisotropic Regularization. I-STAR effectively shapes the geometry of network activations in a stable manner that overcomes the current limitations of other methods that backpropagate through the calculation of principal components during stochastic gradient descent.
3. In contrast to existing theories of NLP, we demonstrate that *decreasing* isotropy in LLMs tends to improve performance on various fine-tuning tasks and models.

## 2 RELATED WORK

**Improving Isotropy.** Methods to restore isotropy in contextualized embedding models fall into two categories: post-processing methods and regularizers. Mu et al. (2017) propose All-But-The-Top, a post-processing algorithm that masks out several of the *top* principal components of the data. The authors show that their simple algorithm improves performance for Word2Vec and GloVe embeddings on word similarity tasks. Several slight variations have occurred on the All-But-The-Top algorithm where the top principal components of the last hidden state of LLMs are masked or removed (Rajaee and Pilehvar, 2021b; Bihani and Rayz, 2021; Liang et al., 2021; Liao et al., 2020; Sajjad et al., 2022). Across each of these studies, the authors argue that improving isotropy in the embedding space improves model performance. However, studies evaluating the impact of improving isotropy in embedding space by masking principal components tend to be limited to word similarity tasks, which do not provide a complete picture of the importance of isotropy in model representations.

Zhou et al. (2020) propose Isotropic Batch Normalization, a modified whitening transform that forces representations to be zero-mean but allows the covariance matrix of model representations to be block diagonal and does not entirely remove all correlations from the data. The authors apply their novel transformation to the final hidden state representations of a BERT model before being input to the classification head and show that Isotopic Batch Normalization minimally improves the performance of BERT on several datasets in the GLUE benchmark. Several authors argue that isotropy can be restored in contextualized embedding space by applying a simple zero-mean transformation to the data (Biś et al., 2021; Cai et al., 2021). Given that isotropy is a property of the covariance matrix of the data and is unrelated to the mean, the improvements on the textual similarity tasks shown in various studies are likely unrelated to the property of isotropy. There have been far fewer attempts in the literature to improve isotropy using regularization penalties. Gao et al. (2019) propose CosReg, a regularization technique that penalizes the model when the average cosine similarity of model representation approaches 1. The motivation behind CosReg is that by reducing the average cosine similarity between embeddings, models will be penalized when representations occupy a "narrow cone" in vector space (Gao et al., 2019; Zhang et al., 2020). Although the authors report modest gains when using CosReg, more current studies have argued that average random cosine similarity does not accurately measure isotropy (Rudman et al., 2022).

Although a large number of papers in NLP argue that isotropy is beneficial for representations, the broader machine learning community has found that 1) anisotropy is a natural consequence of stochastic gradient descent; 2) anisotropy allows for networks to generalize better to unseen examples, and; 3) networks that compress data into lower dimensional manifolds show better performance on downstream tasks (Zhu et al., 2018; Ansuini et al., 2019; Recanatesi et al., 2019). We argue that the primary reason for the differences between claims on isotropy in NLP literature and machine learning literature stems from the noisy range of often flawed methods of measuring isotropy on which many claims are based (Rudman et al., 2022). In Section 2.1, we discuss the most common methods used to measure isotropy in embedding space and detail why most attempts to measure isotropy in the NLP literature do not accurately reflect properties of isotropy.

## 2.1 MEASURING ISOTROPY

**Average random cosine similarity** is the most common method for measuring "isotropy" in embedding space. An average random cosine similarity approaching 1 is thought to represent a minimally isotropic space, while an average random cosine similarity of 0 constitutes a maximally isotropic space (Ethayarajh, 2019). Ethayarajh (2019) finds that the average random cosine similarity between activations of BERT and GPT-2 approach 1, which the authors use to argue that model representations form a "narrow cone" in vector space. However, Rudman et al. (2022) show that average random cosine similarity is not a measure of isotropy since the average cosine similarity of points artificially approaches 1 when the mean of the data is far from the origin and will be 0 when the data are zero-mean, regardless of the shape of the distribution.

The **partition isotropy score** is based on a partition function, $Z(C) := \sum_{x \in X} \exp(c^T x)$, developed by Arora et al. (2015), where $c \in \mathbb{R}^d$ represents the set of unit vectors and $X \subset \mathbb{R}^d$ is a finite point cloud. Since calculating the entire partition function is intractable, studies typically approximate the full partition function as $I(X) \approx \min_{c \in C} Z(c)/\max_{c \in C} Z(c)$, where $c$ is chosen from the eigenspectrum of $XX^T$ (Mu et al., 2017). Mu et al. (2017) prove that there exists a choice of $C$ such that their method for measuring isotropy reflects the uniformity of principal components of the data. However, approximating $c$ from the eigenspectrum of $XX^T$ results in undesirable properties, such as being heavily influenced by the mean vector of $X$ and influenced by orthogonal transformations of the data that are unrelated to isotropy (Rudman et al., 2022).

Intuitively, **IsoScore** measures how "far away" the covariance matrix of the data is from $\alpha \cdot \boldsymbol{I}_d$, where $\alpha$ is a positive scalar and $\boldsymbol{I}_d$ is the $d \times d$ identity matrix. Algorithm 2 details the steps required to calculate IsoScore.Rudman et al. (2022) develop a rigorous testing suite to demonstrate that IsoScore is the first tool in the literature that accurately measures isotropy in embedding space. To ensure that IsoScore is not biased towards a given basis of the data, the authors "reorient" the data by projecting a point cloud of data by its principal components. An IsoScore value of 1 indicates that a distribution is fully isotropic, and a score near 0 suggests that a single dimension in vector space dominates representations. Although IsoScore is an accurate measure of isotropy, we demonstrate in Section 3

---

**Algorithm 1** IsoScore$^\star$ Forward Pass

---

1: **Input**: $X \subset \mathbb{R}^d$ point cloud, $\Sigma_S \in \mathbb{R}^{d \times d}$ shrinkage covariance matrix, $\zeta \in (0, 1)$.
2: **Outputs**: I-STAR penalty of $X$.
3: calculate covariance matrix: $\Sigma_X$ of $X$
4: calculate shrinkage matrix: $\Sigma_\zeta := (1 - \zeta) \cdot \Sigma_X + \zeta \cdot \Sigma_S$
5: calculate eigenvalues: $\Lambda := \{\lambda_1, .., \lambda_d\}$ of $\Sigma_\zeta$
6: normalize eigenvalues: $\hat{\Lambda} := \sqrt{d} \cdot \Lambda / ||\Lambda||_2$ such that $||\hat{\Lambda}|| = \sqrt{d}$
7: calculate the isotropy defect:

$$\delta(\hat{\Lambda}) := ||\hat{\Lambda} - \mathbf{1}|| / \sqrt{2(d - \sqrt{d})}$$

where $\mathbf{1} = (1, ..., 1)^\top \in \mathbb{R}^d$
8: calculate: $\phi(\hat{\Lambda}) := (d - \delta(\hat{\Lambda})^2(d - \sqrt{d}))^2 / d^2$
9: calculate: $\iota(\hat{\Lambda}) := (d \cdot \phi(\hat{\Lambda}) - 1) / (d - 1)$.

---

that IsoScore will systematically underestimate the true isotropy score of a distribution when the number of samples is small relative to the dimensionality of the vector space.

Since many current works in NLP use "flawed" measures of isotropy, such as average random cosine similarity and the partition isotropy score, the connection between isotropy in LLMs and their performance on downstream tasks has not been established. This study devises a novel regularization penalty, I-STAR, to investigate the relationship between model performance and isotropy in model representations. In contrast to several previous works based on average cosine similarity or the partition score, we use IsoScore$\star$ to demonstrate that *decreasing* isotropy tends to improve performance on downstream tasks, while *increasing* isotropy hampers performance on nearly all tasks and models considered in this paper.

## 3 ISOSCORE$^\star$

Previous methods for measuring isotropy in embedding space are either 1) not accurate measures of isotropy, 2) not differentiable, or 3) not stable on mini-batch computations. In this section, we propose IsoScore$^\star$, a novel, fully differentiable measure of isotropy that is stable, even for small sample sizes. First, we thoroughly describe IsoScore$^\star$. We then demonstrate that IsoScore$^\star$ can accurately measure the isotropy of small data samples, while vanilla IsoScore *systematically underestimates* the isotropy of a distribution when the number of points in a finite point cloud is smaller than the dimensionality of the vector space. For the remainder of this section, let $X \subset \mathbb{R}^d$ and $S \subset \mathbb{R}^d$ be finite point clouds drawn from a distribution $\bar{X}$ such that $|X| < d << |S|$.

Intuitively, IsoScore$^\star$ measures the extent to which the principal components of a distribution are uniformly distributed. Measuring isotropy as a function of the principal components allows us to backpropagate through IsoScore$^\star$ since PCA is a differentiable function (Huang et al., 2018). An IsoScore$^\star$ value of 1 implies that for principal components $\{\lambda_1, .., \lambda_d\}, \lambda_i = \lambda_j \forall i, j$. An IsoScore$^\star$ value of 0 implies that only a single principal component is non-zero. Although both IsoScore$^\star$ and the partition score measure isotropy via the principal components of a distribution, IsoScore$^\star$ directly measures the uniformity of the eigenspectrum of principal components and does not have to rely on approximations to the eigenspectrum like the partition function defined by Mu et al. (2017).

**IsoScore$^\star$ Pseudocode.** Step 3) of Algorithm 1 begins by calculating the covariance matrix of $X \subset \mathbb{R}^d$ that we assume is sampled from the distribution $\bar{X}$. Next, in Step 4) we calculate the RDA-Shrinkage matrix, $\Sigma_\zeta$, as a weighted sum between the covariance matrix of $X$ and the covariance matrix of $S$, $\Sigma_S$, to produce a more accurate estimate of the covariance matrix of the true distribution $\bar{X}$. The *shrinkage parameter*, $\zeta$, controls how much covariance information is used from $X$ and $S$ when estimating the covariance matrix of $\bar{X}$. In Step 5), we calculate the eigenvalues of $\Sigma_\zeta$. Step 6) normalizes the eigenvalues so that the L2 norm of the eigenvalues equals the norm of the vector containing all 1s (i.e., $1, 1, ..., 1$). The remaining normalizing Steps 7-9 are derived in the same manner as vanilla IsoScore. For a detailed pseudocode analysis of both IsoScore and IsoScore$^\star$, as well as a proof that IsoScore approximates IsoScore$^\star$ when the number of samples in our point cloud is large, see Section A in the appendix.

**Mini-batch stability of isotropy estimates.** The mini-batch stability of methods measuring isotropy has yet to be investigated. We test the stability of mini-batch isotropy by sub-sampling small batches of points, $X$, from a point cloud of data, $\bar{X} \subset \mathbb{R}^{768}$, consisting of 250,000 points sampled from a 768-dimensional Gaussian distribution with a zero mean-vector and a covariance matrix, $\Sigma_{\bar{X}}$, such that $\Sigma_{\bar{X}}$ has $diag(\Sigma_{\bar{X}}) = \{10, 6, 4, 4, 1, ..., 1\}$ and zero off-diagonal elements.

In Section A of the Appendix, we prove that when $\zeta = 0$ (i.e. no shrinkage is performed), IsoScore($X$) = IsoScore$^\star(X, \zeta, \Sigma_S)$. Figure 2 demonstrates that for a sub-sample $X$ of $\bar{X}$, if $|X|$ is not sufficiently larger than $d$, IsoScore systematically underestimates the true degree of isotropy (dashed horizontal line) of the distribution, $\bar{X}$, from which $X$ is sampled. This means IsoScore($X$) $<<$ IsoScore($\bar{X}$). For isotropy results to be reliable, future work should ensure that the number of points in a given sample is significantly larger than the dimensionality of the distribution. IsoScore underestimates the true degree of isotropy of the distribution because IsoScore relies on calculating the covariance matrix of a sample. When the number of points in a sample, $|X|$, is less than the dimensionality of the space, the covariance matrix of $X$ may be singular (Friedman, 1989). Existing methods to improve isotropy and some of the most common metrics to evaluate isotropy, such as the partition isotropy score (Mu et al., 2017), rely on investigating the principal components of the data. As a consequence, the problem of underestimating the isotropy of sample distributions will affect nearly all previous works. Fortunately, many well-established methods in the statistics and machine learning literature exist to ensure that a covariance matrix will be better conditioned and invertible, leading to more reliable methods to alter and measure isotropy in embedding space (Friedman, 1989).

**Stabilizing covariance estimation.** Shrinkage is a simple operation that adds a known, stable covariance matrix to a non-invertible, singular sample covariance matrix (Friedman, 1989). Performing shrinkage ensures that the resulting covariance matrix is invertible. In situations where one does not have access to multiple samples or a sample where $S \subset \bar{X}$ such that $d << |S|$, the matrix $\zeta \cdot I_d + \Sigma_X$ is used as the shrinkage matrix (Friedman, 1989). However, if one has access to the larger point cloud $S$ or multiple samples from $\bar{X}$, then a more faithful estimate of the covariance matrix of $\bar{X}$ can be obtained using regularized discriminant analysis (RDA) (Friedman, 1989). RDA shrinkage pools covariance matrices together using $\Sigma_\zeta := \zeta \cdot \Sigma_X + (1 - \zeta) \cdot \Sigma_S$ to get a more accurate measure of the covariance matrix of the distribution from which $X$ is sampled. Figure 2 demonstrates that performing this shrinkage operation on $\Sigma_X$

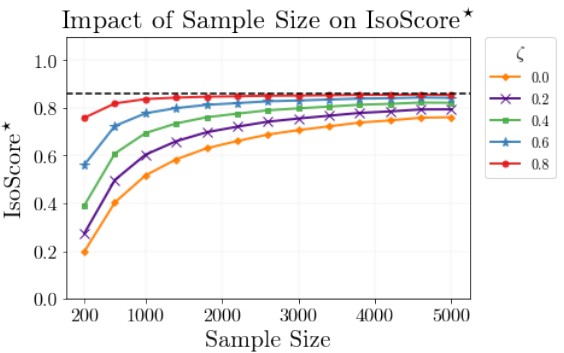

Figure 2: IsoScore$^\star(X, \zeta, \Sigma_S)$ values for different choices of $\zeta$. The dashed line indicates the correct IsoScore$^\star$ value of $\bar{X}$, which is IsoScore$^\star(\bar{X}) = 0.86$. We calculate $\Sigma_S$ from a subsample $S \subset \bar{X}$ such that $X \cap S = \emptyset$ and $|S| = 75,000$.

drastically improves the stability of IsoScore$^\star$ even when $|X| = 700$ and $d = 768$. Step 4 of Algorithm 1 uses RDA shrinkage to stabilize IsoScore$^\star$ on mini-batches during training. In stochastic gradient descent, mini-batches of data are sampled randomly from the larger training set. We perform shrinkage on the mini-batch covariance matrix with a covariance matrix calculated from token embeddings obtained from a small fraction of the training data to obtain the most accurate estimate of mini-batch isotropy. We initialize the shrinkage matrix, $\Sigma_S$, by computing the covariance matrix of a sample, $S$, of 250k points obtained from running a partial forward pass on the training data before training with I-STAR. We update $\Sigma_S$ after each epoch during training by running a partial forward pass on the training data. We use IsoScore$^\star$ as the penalty in our loss function to produce stable updates during training that alter the isotropy of model representations. Figure 1 illustrates how we incorporate IsoScore$^\star$ to form our I-STAR loss. We take the union of token embeddings from each hidden state in the model and calculate a global IsoScore$^\star$ penalty to incorporate into our loss function. We stress that shrinkage is crucial for the success of I-STAR. In Section D, we show that *without shrinkage*, the performance of models tuned with I-STAR can drop by as much as $\approx 6\%$.

## 4 METHODS

**CosReg.** To compare CosReg to I-STAR, we adapt the cosine similarity regularization term presented by (Gao et al., 2019) and calculate our CosReg penalty on the last-layer hidden states of the model. Let $\{x_1, x_2, ..., x_M\}$ denote the mini-batch representation obtained from the last hidden layer, $X_n$, of a contextualized embedding model, let $\hat{x}_i = \frac{x_i}{||x_i||}$ and let $\lambda \in \mathbb{R}$ be a tuning parameter, then the CosReg loss function of our model is defined as:

$$L_{\text{CosReg}} = L_{\text{CE}} + \lambda \frac{1}{M^2} \sum_i^M \sum_{j \neq i} \hat{x}_i^T \hat{x}_j \tag{1}$$

We use $\lambda = 1$ as Gao et al. (2019) find that using $\lambda = 1$ is sufficient for altering average random cosine similarity and that using $\lambda > 1$ does not provide any additional training benefits. Since an average cosine similarity of 1 is thought to reflect an anisotropic space and an average cosine similarity near 0 reflects an isotropic space, using $\lambda = 1$ is believed to encourage "isotropy" as measured by cosine similarity (Ethayarajh, 2019). However, we show in Figure 4 that CosReg impacts the mean of embeddings but has no impact on isotropy. Namely, *CosReg does not properly regularize isotropy*.

**I-STAR.** Figure 1 outlines the calculation of I-STAR loss. I-STAR computes a global IsoScore$^\star$ penalty from token embeddings obtained from all layers in the network. Calculating a global isotropy penalty from the representations at every layer of the network allows the model to determine where changing the isotropy of representations in the network will lead to the largest improvements in performance. In Section G in the Appendix, we examine the impact of applying I-STAR to individual layers and find that calculating a global isotropy penalty leads to the most consistent performance. Let $\tilde{X} = \bigcup_{l=1}^n X_l$ denote the union of all hidden states from a network with $n$ layers and $\Sigma_{S_i}$ be the shrinkage covariance matrix for epoch $i$ of training. We define our I-STAR loss as follows:

$$L_{\text{I-STAR}} = L_{\text{CE}} + \lambda \cdot (1 - \text{IsoScore}^\star(\tilde{X}, \zeta, \Sigma_{S_i})) \tag{2}$$

A negative value of $\lambda$ will decrease the levels of isotropy in model representations, while positive choices of $\lambda$ will increase levels of isotropy.

**Intrinsic dimensionality estimation.** Previous studies have shown that compressing the intrinsic dimension of LLM activations in later layers is correlated with improved performance on downstream tasks (Recanatesi et al., 2019). To determine the relationship between isotropy and intrinsic dimensionality, we use TwoNN to estimate the intrinsic dimension of activations similarly to (Recanatesi et al., 2019). TwoNN is a widely used intrinsic dimensionality (ID) estimator based on the ratio between each point's first and second nearest neighbors (Facco et al., 2017). Note that calculating TwoNN is an iterative process based on the pairwise distances of nearest neighbors in space and is a non-differentiable operation.

### 4.1 EXPERIMENTAL DESIGN

**Models & datasets.** In this paper, we fine-tune BAppendixepbert, ALBERT (Lan et al., 2020), and DistilBERT (Sanh et al., 2020) for nine NLP common benchmark tasks: SST-2 (Socher et al., 2013), QNLI (Rajpurkar et al., 2016a), RTE (Dagan et al., 2005), MRPC (Dolan and Brockett, 2005), QQP (Wang et al., 2018), COLA (Warstadt et al., 2019), STS-B (Cer et al., 2017), SST-5 Socher et al. (2013) and SQUAD Rajpurkar et al. (2016b). A detailed description of each dataset is available in Section F

**Hyperparameters & training details.** For each model and each task, we hyperparameter tune for batch size (8, 16, 32), training epochs (3,4,5), and learning rate ($1e$-5, $3e$-5, $5e$-5). For I-STAR, we tune for the optimal zeta (0.2, 0.4, 0.6, 0.8) and use the tuning parameters values, $\lambda \in \{-5, -3, -1, 1, 3, 5\}$. For CosReg, we use a tuning parameter of 1 in accordance with Gao et al. (2019). All reported performance metrics are calculated as an average over five random seeds to demonstrate the robustness of our results. After we perform our hyperparameter tuning, we fine-tune our models using two 3090-RTX GPUs, use mixed-point precision training for all models/tasks, and set a gradient accumulation step to 2.

## 5 RESULTS

In this section, we demonstrate that 1) there is an *inverse* relationship between isotropy and performance; 2) CosReg implements a zero mean transform and does not improve isotropy, and 3) increasing/decreasing isotropy using I-STAR increases/decreases the TwoNN intrinsic dimensionality estimation of model representations.

Table 1: Performance of CosReg and I-Star for each model and task. "Base" indicates that no regularization methods were used. For COLA, we report Matthew's Correlation; for STS-B, we report Pearson's Correlation; for SQUAD, we present F1/EM. For all remaining tasks, we report accuracy. We report the average/standard deviation over 5 random seeds. Each I-STAR value comes from training with a **negative** tuning parameter. See Table 2 for more details.

| Method | SST-2 | QNLI | RTE | MRPC | QQP | COLA | STS-B | SST-5 | SQUAD |
|---|---|---|---|---|---|---|---|---|---|
| ALBERT I-STAR | **93.08±0.32** | **91.41±0.43** | **72.56±1.29** | **87.84±0.43** | 89.97±0.06 | **55.22±1.61** | 88.91±0.23 | **55.02±0.46** | **90.33±0.09/82.96±0.04** |
| ALBERT Base | 93.08±0.24 | 90.28±0.42 | 71.34±0.81 | 86.96±0.32 | **89.98±0.06** | 54.56±0.93 | **89.07±0.20** | 54.23±0.30 | 90.32±0.13/82.86±0.17 |
| ALBERT CosReg | 91.86±0.42 | 90.91±0.25 | 65.99±1.81 | 86.91±0.26 | 89.81±0.14 | 49.59±0.90 | 87.98±0.46 | 54.76±0.11 | 90.08±0.31/82.64±0.32 |
| BERT I-STAR | **92.65±0.18** | 89.51±0.69 | **62.53±0.70** | **86.76±0.57** | 90.36±0.05 | **59.60±0.25** | **86.44±0.16** | **50.64±0.34** | **87.76±0.11/80.02±0.10** |
| BERT Base | 92.40±0.41 | 89.49±0.30 | 62.24±0.48 | 86.51±0.21 | 90.44±0.11 | 59.22±0.63 | 86.15±0.20 | 50.00±0.28 | 87.02±0.11/78.82±0.25 |
| BERT CosReg | 92.01±0.23 | **89.67±0.56** | 61.52±1.41 | 85.29±0.47 | **90.45±0.82** | 57.90±0.82 | 86.30±0.28 | 49.44±0.18 | 86.89± 0.14/78.85±0.08 |
| DistilBERT I-STAR | **91.42±0.11** | 86.25±0.32 | **58.05±0.95** | **84.17±0.35** | **89.67±0.08** | 50.03±0.93 | **85.28±0.11** | **50.13±0.23** | **83.69±0.36/74.80±0.27** |
| DistilBERT Base | 91.36±0.15 | **87.40±0.34** | 56.82±0.67 | 83.68±0.41 | 89.57±0.74 | **50.16±0.59** | 84.75±0.26 | 49.48±0.28 | 83.23±0.18/74.04±0.12 |
| DistilBERT CosReg | 90.94±0.38 | 86.68±0.19 | 57.04±1.13 | 83.58±0.44 | 89.55±0.08 | 49.30±0.88 | 84.41±0.08 | 48.65±0.25 | 83.47± 0.34/74.40±0.24 |

**There is an inverse relationship between isotropy and performance.** In contrast to previous studies, Figure 3 demonstrates that increasing isotropy tends to decrease performance, whereas decreasing isotropy tends to increase performance. We observe that the trend's strength is somewhat task and model-dependent, with the most robust correlation emerging for SQUAD, RTE, and STS-B. However, for both MRPC and COLA, ALBERT does not exhibit a strong correlation between isotropy and performance. However, for both bases, the optimal tuning parameter is negative. The lack of a distinct trend for ALBERT on MRPC and COLA is likely due to the large amounts of noise in fine-tuning performance and isotropy values occurring on these tasks. Further, Table 1 demonstrates that decreasing isotropy using negative tuning parameters in I-STAR improves performance over both baseline models and CosReg on most tasks and models considered in this paper.

**CosReg implements a zero-mean transform.** Figure 4 demonstrates that CosReg alters the mean of model activations, particularly in a single dimension. Encouraging the average random cosine similarity of activations to be 0 (i.e., when $\lambda = 1$) forces representation to be zero-mean, whereas encouraging an average random cosine similarity to be 1 (i.e., when $\lambda = -1$) causes the mean of representations to increase further. Although CosReg impacts the mean of the data, CosReg does not increase isotropy in model representations. After fine-tuning BERT, ALBERT, and DistilBERT on SST-2 using CosReg with a tuning-parameter value of $\lambda = 1$, the last layer representations of each model receive IsoScore* values of 0.004, 0.007, and 0.007, respectively.

**Isotropy and intrinsic dimensionality estimation.** Several studies have found that a lower intrinsic dimension of later layer representations is correlated with improved performance on various downstream tasks (Ansuini et al., 2019; Recanatesi et al., 2019). Figure 5 demonstrates that adjusting isotropy with I-STAR corresponds to changing the intrinsic dimension of model representations. Importantly, encouraging isotropy in model representations does not allow for model representations to compress into a lower dimensional manifold in later layers.

## 6 DISCUSSION

Our study challenges a dominant belief in the NLP literature that encouraging isotropy improves performance on downstream tasks. In contrast to several previous works, we find that encouraging isotropy is detrimental to model performance and that *decreasing* isotropy in representations improves performance on a broad range of tasks and models. Table 1 and Figure 3 provide strong empirical evidence, in support of (Zhu et al., 2018), that anisotropy is essential for a model's downstream performance.

The primary reason for the discrepancy between our results and existing studies in the NLP literature on isotropy is that previous studies have made claims using "flawed" measures of isotropy, such as average random cosine similarity. Figure 4 shows that using CosReg implements a zero-mean transform and *does not improve isotropy*. Given our findings that isotropy and classification performance

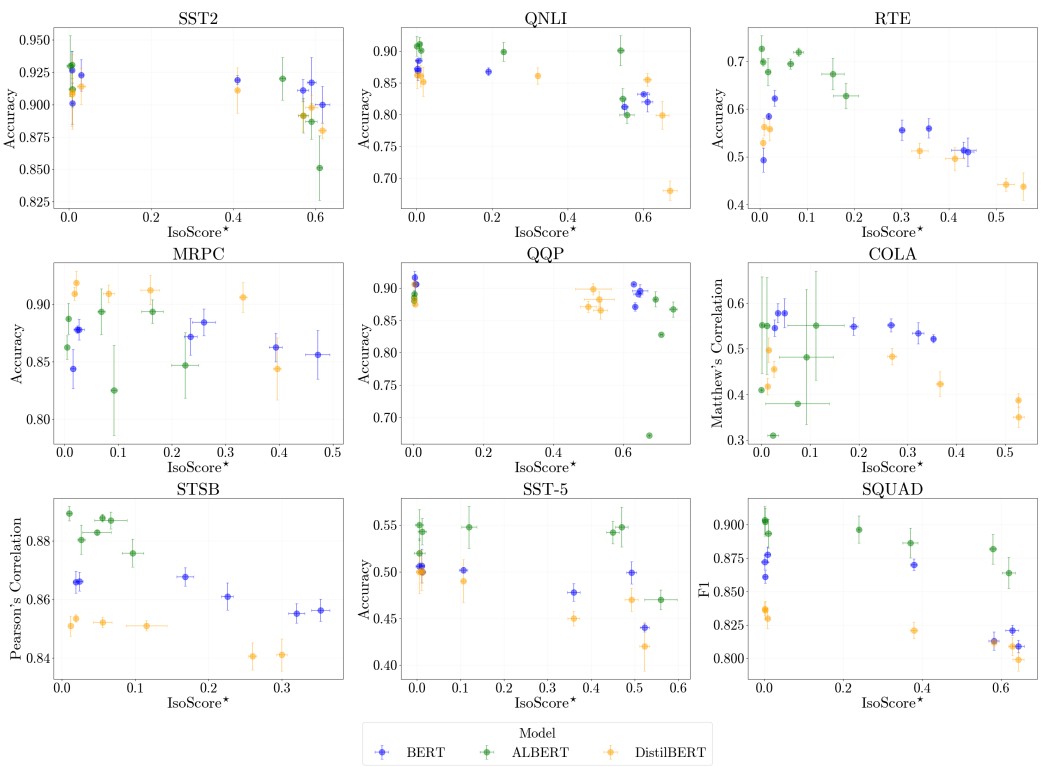

Figure 3: Relationship between IsoScore* (x-axis) and model performance (y-axis). We fine-tune each model with I-STAR using the tuning parameters $\lambda \in \{$-5, -3, -1, 0.50, 1, 3, 5$\}$. We train each model over five random seeds and report the standard deviation of both performance and IsoScore$^\star(X, \zeta, \Sigma_S)$ values. We set $\zeta = 0.2$ for all computations of IsoScore$^\star$, and we compute $\Sigma_S$ from a random sample of 250,000 token embeddings from the training data.

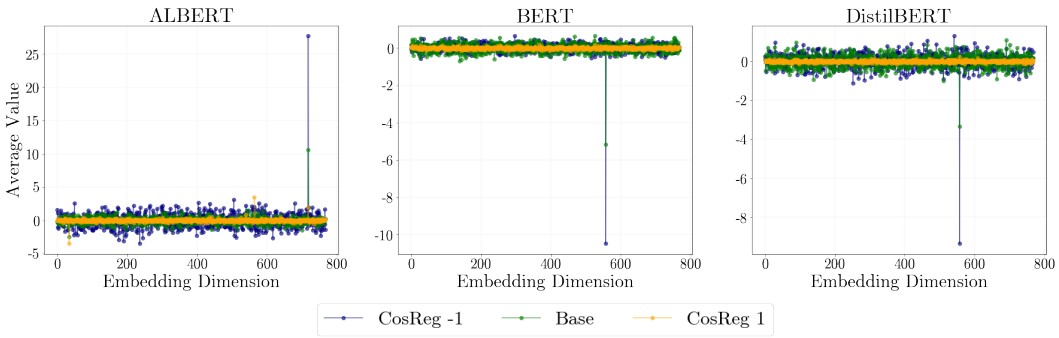

Figure 4: Comparing the mean activation values on the validation data for each dimension of ALBERT, BERT, and DistilBERT fine-tuned on QNLI, with CosReg using a tuning-parameter value of $\lambda = -1, 1$ and without any regularization. Trends are representative of all tasks.

are negatively correlated and that CosReg does not adjust isotropy, we argue many of the current claims regarding isotropy in NLP need to be reassessed.

Although a majority of prior studies in NLP have argued that isotropy is beneficial to LLMs, recent works have helped to explain our finding that isotropy negatively correlates with classification performance. (Mickus et al., 2024) provide a robust mathematical framework demonstrating that

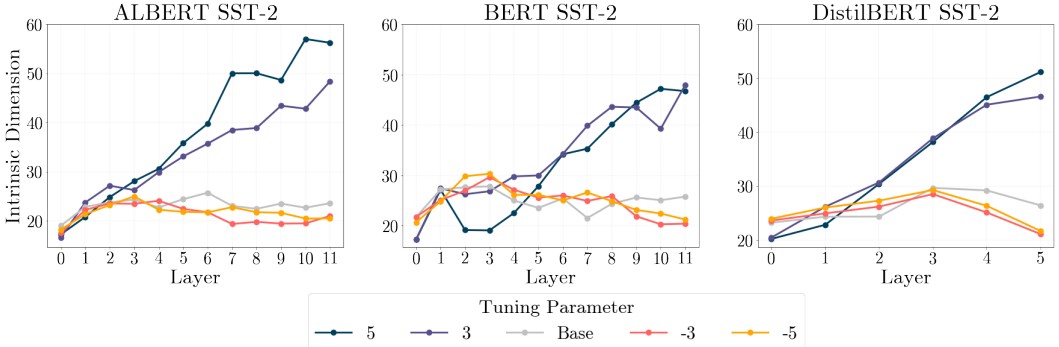

Figure 5: TwoNN Intrinsic Dimensionality estimate of ALBERT, BERT, and DistilBERT sentence embeddings, i.e. [CLS] tokens, obtained from the SST-2 validation data for models fine-tuned on the SST-2 using I-STAR with tuning-parameters $\lambda \in \{-5, -3, 3, 5\}$. "Base" represents the case where no regularization is used. Trends are representative of all tasks.

isotropy and clustering are incompatible objectives and that clustering behavior is crucial for an effective classifier. Namely, encouraging isotropy inhibits clustering behavior, which is harmful to downstream performance.

Our findings strongly support arguments in the literature outside of NLP that anisotropy is a natural outcome of stochastic gradient descent and that compressing representations is necessary for model performance. Additionally, studies have shown that model representations that occupy a lower intrinsic dimension in the ambient vector space tend to outperform those sampled from higher dimensional manifolds (Recanatesi et al., 2019; Ansuini et al., 2019). We demonstrate that encouraging isotropy in the embedding space increases the intrinsic dimension of model representations, which is detrimental to performance. Importantly, we also show that reducing isotropy in the embedding space leads to the compression of representations into a lower dimensional manifold, resulting in improved model performance. This underscores the critical role of isotropy in determining the intrinsic dimension of model representations and the subsequent impact on model performance.

**Limitations.** Although having isotropic representations is theoretically desirable for both the interpretability of model decisions and for improved quantization abilities, encouraging isotropy in pre-trained models in a way that preserves or improves downstream task performance is challenging. This study is limited to fine-tuning LLMs, which may not provide a definitive answer to whether encouraging isotropy in embedding space is inherently detrimental to model performance. Our results demonstrate that encouraging isotropy in pre-trained models causes a decline in downstream fine-tuning performance. Fine-tuning requires models to make rapid and drastic adjustments to their representations within a limited number of training steps. A fruitful direction for future work would consist of using I-STAR in LLM pre-training to enforce isotropic representations throughout training.

## 7 CONCLUSION

Previous works in NLP have argued that anisotropy in contextualized embedding models limits the expressiveness of word representations and forces embeddings to occupy a "narrow cone" in vector space. Several studies have claimed that improving isotropy leads to improved performance on downstream tasks. However, most studies use faulty isotropy measures, tend to be limited to word similarity tasks, and only investigate isotropy for last-layer representations. We propose I-STAR, a differentiable, mini-batch-stable isotropy-based regularization scheme, to study the relationship between fine-tuned model performance and isotropy. Contrary to previous works in NLP, we find that further *decreasing* isotropy improves downstream model performance. Fundamentally, we show that enhancing isotropy in embedding space increases the intrinsic dimensionality of model representations and causes model performance to decrease. Given the connection between isotropy, intrinsic dimensionality, and performance, I-STAR shows great promise for application in various areas of deep learning.

## 8 REPRODUCIBILITY

We have taken several steps to make our paper as reproducible as possible. Firstly, we have made all code used to produce the project publicly available and attached an anonymous version along with our submission. Further, we have released a pip install of IsoScore$^\star$ to facilitate future works. Section G outlines all values of the two critical hyperparameters needed to train with I-STAR loss. Lastly, all datasets and models used in this paper are publicly available on Huggingface.

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

---

**Algorithm 2** IsoScore

---

1: **Input**: Let $X \subset \mathbb{R}^d$ be a finite collection of points.
2: Let $X^{\mathrm{PCA}}$ denote the points in $X$ transformed by the first $n$ principal components.
3: Define $\Sigma_D \in \mathbb{R}^n$ as the diagonal of the covariance matrix of $X^{\mathrm{PCA}}$.
4: Normalize diagonal to $\hat{\Sigma}_D := \sqrt{n} \cdot \Sigma_D / \|\Sigma_D\|$, where $\| \cdot \|$ is the standard Euclidean norm.
5: The isotropy defect is

$$\delta(X) := \|\hat{\Sigma}_D - \mathbf{1}\| / \sqrt{2(n - \sqrt{n})}$$

   where $\mathbf{1} = (1, \ldots, 1)^\top \in \mathbb{R}^n$.
6: $X$ uniformly occupies

$$\phi(X) := (n - \delta(X)^2 (n - \sqrt{n}))^2 / n^2$$

   percent of ambient dimensions.
7: Transform $\phi(X)$ so it can take values in $[0, 1]$, via $\iota(X) := (n \cdot \phi(X) - 1)/(n - 1)$.
8: **return:** $\iota(X)$

---

## A   ISOSCORE VS ISOSCORE$^\star$

**IsoScore**$(X)$ **= IsoScore**$^\star(X, \zeta, C_X)$ **when** $\zeta = 0$. Let $X \subset \mathbb{R}^n$ be a finite point cloud that we assume is sampled from some larger distribution $\bar{X}$ such that the number of points in $\bar{X}$ is sufficiently larger than the number of points in $X$. Let $\zeta \in (0, 1)$ be a shrinkage parameter, and let $\Sigma_S \in \mathbb{R}^{n \times n}$ be a shrinkage covariance matrix obtained from a sample of points, $S$, drawn from $\bar{X}$ such that $|S| >> n$. We will first demonstrate that without RDA-Shrinkage (i.e. when $\zeta = 0$), IsoScore$(X)$ = IsoScore$^\star(X, \zeta, \Sigma_S)$. The key insight is that $\Sigma_D$ in Step 3 of Algorithm 1 is equivalent to the principal components of $X$.

Algorithm 2 shows that the first step of IsoScore is to transform $X$ to its principal components to get what the authors denote as $X_{\mathrm{PCA}}$. Let $\Sigma_{\mathrm{PCA}}$ be the covariance matrix of $X_{\mathrm{PCA}}$, and let $\Sigma_X$ denote the covariance matrix of $X$. Projecting $X$ to its principal components removes correlations from the data, meaning that $\Sigma_{\mathrm{PCA}}$ will be diagonal. Since $\Sigma_{\mathrm{PCA}}$ is a diagonal matrix, its eigenvalues are equal to $diag(\Sigma_{\mathrm{PCA}})$. Therefore, $diag(\Sigma_{\mathrm{PCA}})$ are the principal components of $X_{PCA}$ since principal components are the eigenvalues of the covariance matrix. Note that principal components are invariant under orthogonal transformations and that "reorienting" the data by PCA applies the orthogonal transformation $V^T X V$, where $V$ are the eigenvectors of $\Sigma_X$. Namely, the principal components of $X$ are the principal components of $X_{PCA}$. For a simple proof demonstrating that the principal components of $X$ are invariant under any orthogonal transformation applied to $X$, see (Ding et al., 2006). Therefore, IsoScore$^\star$ is equivalent to IsoScore when no covariance matrix shrinkage is performed. That is, when $\zeta = 0$, IsoScore$(X) =$ IsoScore$^\star(X, \zeta, \Sigma_S)$ $\forall X \in \mathbb{R}^N$. To see that IsoScore$(X)$ approaches IsoScore$^\star(X, \zeta, \Sigma_S)$, we can use the Law of Large Numbers to show that the larger the sample of $X$, the more close $X$ approximates the true distribution $\hat{X}$. Therefore, IsoScore$(X) \to$ IsoScore$^\star(X, \zeta, \Sigma_S)$ as $|X|$ increases.

**Comparing Pseudocode.** IsoScore$^\star$ addresses two fundamental flaws in vanilla IsoScore. Firstly, IsoScore$^\star$ uses *RDA-shrinkage* (Friedman, 1989) to stabilize the calculation of a sample covariance matrix. Secondly, IsoScore$^\star$ removed all non-differentiable operations present in IsoScore. The pseudocode for IsoScore and IsoScore$^\star$ has two primary steps: 1) extract distribution/isotropy information from the point cloud, $S$ and 2) normalize the isotropy information into a score in the closed interval $[0, 1]$. In Algorithm 3, **Steps 3-4** calculate the covariance matrix of the point cloud $S$, and perform RDA-shrinkage by taking the weighted sum of the covariance matrix of $S$ and a covariance matrix of a larger distribution from which $S$ was sampled. **Step 5** then calculates the eigenvalues from the resulting sum of covariance matrices. When we set $\zeta$ in **Step 4** to 0, the resulting eigenvalues are the principal components of our sample $S$. All normalizing steps are identical in IsoScore and IsoScore$^\star$. Namely, lines 6-9 in Algorithm 1 are equivalent to lines 4-7 in Algorithm 2.

**Non-Differentiability of IsoScore.** We want to highlight the exact step in Algorithm 2 that makes IsoScore non-differentiable. Step 3 in Algorithm 2 involves selecting the diagonal of a covariance matrix, which is a non-differentiable operation. Note that the non-differentiability of IsoScore does *not* imply that IsoScore$^\star$ is non-differentiable, even though IsoScore and IsoScore$^\star$ are equivalent when no shrinkage is performed. To further emphasize this point, consider the two functions $f(x) = x \forall x$

and $g(x) = x$ for all $x! = 0$ and $g(x) = |x|$ when $x = 0$. Here, $|x|$ indicates the absolute value function. For all inputs $x$, $f(x) = g(x)$. However, $f(x)$ is differentiable for all values of $x$, where $g(x)$ is *not* differentiable when $x = 0$. Section 3 demonstrates that IsoScore$^\star$ is equivalent to IsoScore when no shrinkage is performed. IsoScore$^\star$ preserves all of the desirable theoretical properties of IsoScore while improving stability on small inputs and removing the operation that makes IsoScore non-differentiable (namely, an iterative selection of the diagonal).

# B  WHY DO MODELS PREFER ANISOTROPY?

# C  LAYER-WISE ISOTROPY

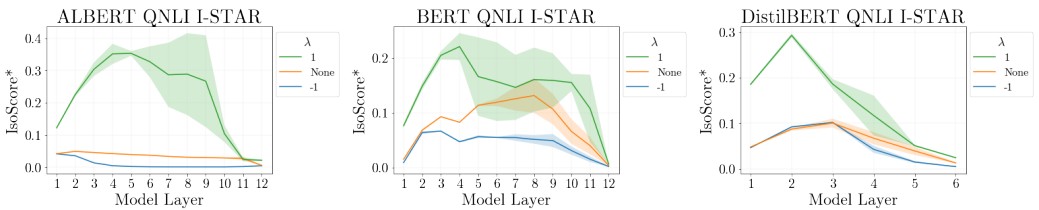

Figure 6: Layer-wise IsoScore$^\star$ values for ALBERT, BERT, and DistilBERT fine-tuned with I-STAR using tuning parameters $\lambda \in \{-1, 1\}$. IsoScore$^\star$ values are calculated on the QNLI validation data using a shrinkage parameter of $\zeta = 0.2$. "None" indicates that no regularizer is used in fine-tuning.

When fine-tuning models with I-STAR, we compute the IsoScore$^\star$ penalty from the union of all token embeddings from each model layer. This section analyzes what layers in the model are impacted the most by our I-STAR regularizer.

Figure 6 shows that encouraging isotropy in token embeddings using I-STAR primarily impacts early layers in the network. Even when isotropy is encouraged using positive tuning parameter values in I-STAR, token representations from the later layers of the network remain highly anisotropy. These results provide further evidence that anisotropy in the form of outlier dimensions that emerge in the last layers of the network is crucial to the model decision-making process. An interesting direction for future work could be to explore applying I-STAR to various layers in the network.

# D  IMPACT OF THE SHRINKAGE PARAMETER ON I-STAR

In this section, we evaluate the impact of changing the shrinkage parameter, $\zeta$, on downstream performance when fine-tuning with I-STAR regularization. To test the effect of varying $\zeta$, we fix all hyperparameters found in Table 2 and fine-tune with $\zeta \in \{0, 0.2, 0.4, 0.6, 0.8, 1\}$. We train ALBERT, BERT, and DistilBERT on RTE. All results are reported as an average over 5 random seeds.

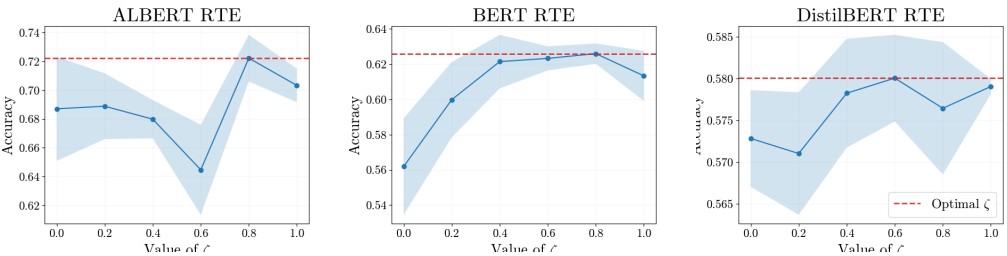

Figure 7: Performance of ALBERT, BERT, and DistilBERT on RTE with changing values of $\zeta$. The red dashed line denotes the optimal value of $\zeta$. Setting $\zeta = 0$ indicates that no shrinkage is performed (i.e., equivalent to regularizing using IsoScore), and setting $\zeta = 1$ signifies that no mini-batch covariance information is included in the gradient updates during a backward pass.

In Section 3, we demonstrated that IsoScore systematically underestimates the actual value of point clouds when the number of samples is lower than the dimensionality of the vector space. IsoScore$^\star$ overcomes this fundamental limitation by using *shrinkage* to improve the stability of the covariance matrix calculation. Figure 7 demonstrates the importance of using IsoScore$^\star$ when computing isotropy scores of a mini-batch. When $\zeta = 0$ and no shrinkage is performed, the performance of our fine-tuned model decreases by $3.85\%, 6.19\%$, and $0.76\%$ compared to optimal values of $\zeta$ for ALBERT, BERT, and DistilBERT, respectively.

In addition to testing the utility of using covariance matrix shrinkage, test the impact of excluding mini-batch covariance information by setting $\zeta = 1$. When $\zeta = 1$, IsoScore$^\star$ is calculated from the stabilizing covariance matrix, $\Sigma_{S_i}$, obtained by calculating the covariance matrix from a sample of 250,000 points at epoch $i$. Figure 7 demonstrates the utility of using mini-batch covariance matrix information during fine-tuning as the optimal tuning parameter is always a value of $\zeta \in (0, 1)$.

## E  APPLYING I-STAR TO DIFFERENT LAYERS

Throughout this paper, we calculate a *global* isotropy penalty on the vector space of all token embeddings from all layers in the model. Our motivation for selecting a global isotropy penalty instead of penalizing individual layers is two-fold. Firstly, IsoScore$^\star$ is a more faithful representation of the true isotropy of the global vector space when the number of samples is large, and the sample covariance is more likely to be full-rank, meaning that isotropy-based gradient updates will be more stable. Secondly, selecting individual layers to penalize would drastically increase the hyperparameter search when using I-STAR. Lastly, we ultimately decided on a global isotropy over regularizing individual layers as a global isotropy penalty resulted in the most consistent results across different tasks.

In this section, we fine-tune ALBERT, BERT, and DistilBERT on COLA using 5 random seeds. We fix all hyperparameters in Table 2 except for the layer we select to apply I-STAR.

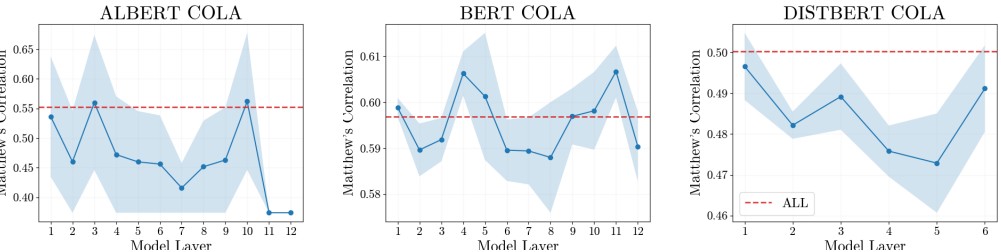

Figure 8: Impact of applying I-STAR on different layers in ALBERT, BERT, and DistilBERT fine-tuned on COLA. The dashed red line is the result of applying a global I-STAR penalty on all layers of the model.

Figure 8 reports the performance of applying I-STAR to individual layers in the network. Although regularizing individual layers in ALBERT and BERT can improve performance compared to a global isotropy penalty, a global isotropy penalty provides much more consistent performance across models and tasks. Namely, no consistent patterns emerge when applying I-STAR to individual layers. However, more work is needed to determine if the occasional improvement in performance is worth the extensive hyperparameter search induced by layer-wise I-STAR.

## F  DATASET DETAILS

Stanford Sentiment Treebank with 2 classes (SST-2) is a binary classification task where models must determine whether a short movie review is positive or negative in sentiment (Socher et al., 2013). SST-5 is a five-class version of SST-2 where the models must determine whether a movie review is negative, somewhat negative, neutral, or positive. QNLI is a binary natural language inference task where models must decide whether or not a given answer is entailed from a specified question (Wang et al., 2018). Stanford Question Answering Dataset V1 (SQUAD)is an extractive question-answering task where a model must select the span of text in a passage that answers a given question

(Rajpurkar et al., 2016b). Recognizing Textual Entailment (RTE) is a binary classification task where a model must determine if a given sentence logically follows a preceding sentence. STS-B (Semantic Textual Similarity Benchmark) is a collection of sentence pairs annotated with a similarity score from 1-5. STS-B is commonly evaluated with Pearson's correlation coefficient. The Microsoft Research Paraphrase Corpus (MRPC) tasks models with determining if a pair of sentence are paraphrases of each other (i.e. semantically equivalent). Quora Question Pairs (QQP) consist of question pairs from Quora. Models must determine if the sentence pairs are semantically equivalent. Corpus of Linguistic Acceptability (COLA) task models to determine if a given string is a linguistically acceptable sentence. SST-2, QNLI, RTE, MRPC, STS-B, QQP, and MRPC are all datasets in the GLUE benchmark (Wang et al., 2018).

## G  I-STAR HYPERPARAMETERS

Table 2: Optimal I-STAR hyperparameter values of the tuning parameter, $\lambda$ and shrinkage parameter, $\zeta$. We searched over $\lambda \in \{-5, -3, -1, 1, 3, 5\}$ and $\zeta \in \{0.2, 0.4, 0.6, 0.8\}$. We present results as $\lambda \mid \zeta$. Note that all values of $\lambda$ are negative, meaning I-STAR will *decrease* isotropy in embedding space.

| Method | SST-2 | QNLI | RTE | MRPC | QQP | COLA | STS-B | SST-5 | SQUAD | Avg. |
|---|---|---|---|---|---|---|---|---|---|---|
| ALBERT I-STAR | -1 \| 0.2 | -1 \| 0.2 | -1 \| 0.8 | -1 \| 0.4 | -1 \| 0.2 | -1 \| 0.8 | -1 \| 0.4 | -1 \| 0.2 | -5 \| 0.2 | **-1.44 \| 0.36** |
| BERT I-STAR | -5 \| 0.2 | -1 \| 0.4 | -1 \| 0.8 | -1 \| 0.6 | -1 \| 0.2 | -1 \| 0.8 | -1 \| 0.2 | -5 \| 0.2 | -3 \| 0.6 | **-2.11 \| 0.44** |
| DistilBERT I-STAR | -5 \| 0.6 | -1 \| 0.2 | -1 \| 0.6 | -1 \| 0.8 | -1 \| 0.2 | -3 \| 0.4 | -3 \| 0.4 | -1 \| 0.2 | -5 \| 0.2 | **-2.33 \| 0.40** |

In this Section, we outline the optimal hyperparameters used in I-STAR for each task and each model. I-STAR requires two key hyperparameters: $\lambda$, the tuning parameter, and $\zeta$, the shrinkage parameter. Recall that the tuning parameter controls both the strength of the signal of the IsoScore$^\star$ in the loss function and whether isotropy is encouraged or discouraged in model representations. Namely, when $\lambda > 0$, I-STAR will increase isotropy in the embedding space, and when $\lambda < 0$, I-STAR will decrease isotropy in the embedding space. The shrinkage parameter, $\zeta$, determines how much covariance information comes from a sample point cloud, $X$, and how much covariance information comes from $C_0$. For all tasks and models, we limit our hyperparameter search to $\lambda \in \{-5, -3, -1, 1, 3, 5\}$ and $\zeta \in \{0.2, 0.4, 0.6, 0.8\}$. *Note that all optimal tuning parameter values occur when $\lambda < 0$, meaning further decreasing isotropy leads to better performance gains than encouraging isotropy.*

