# OpenReview forum: "Stable Anisotropic Regularization"
_ICLR.cc/2024/Conference — ICLR 2024 poster_

### Official Review · Reviewer_tWNn · 2023-11-01

**Soundness:** 2 fair
**Presentation:** 3 good
**Contribution:** 2 fair
**Rating:** 3
**Confidence:** 4

**Summary:**

The authors improved upon IsoScore, a measure for the isotropy of point clouds (Findings of ACL'22), and proposed the IsoScore* as a measure of **an**isotropy. IsoScore* employs PCA (as a natural method for estimating the covariance matrix) and utilizes shrinkage estimation, making it a differentiable measure that operates robustly even with small data sizes. In experiments, IsoScore* was used as a regularization term when fine-tuning masked language models. As a result, under optimal hyperparameters, performance improvements were observed across multiple downstream tasks.

**Strengths:**

- In the NLP field, the isotropy of internal representations was believed to be the key to model success. The message of this paper, suggesting that the **an**isotropy of internal representations might be the key to performance improvement, will likely resonate intriguingly with many readers.
- The paper comprehensively covers a collection of works related to the isotropy of NLP models, making it a highly self-contained piece for readers.

**Weaknesses:**

### 1. The reasons for contradictions with prior research are unclear, weakening the persuasiveness of the main claim.
The authors' main claim that "**an**isotropy is the key to model performance improvement" isn't reconciled with prior research which posits that "isotropy is the key to model performance improvement". While the authors suggest that the discrepancy arises from the evaluation metrics used (as stated “previous studies have made claims using “flawed” measures of isotropy,” on page 7), both this work and prior studies differ *not* only in metrics but also in tasks (GLUE tasks vs. word similarity tasks). Therefore, it's not an apple-to-apple comparison. If there's an inverse correlation between anisotropy and performance in word similarity tasks, it becomes challenging to coherently explain the overall results. This issue might possibly be resolved if there were experiments or comprehensive discussions specifically for the word similarity tasks.

### 2. The claim that the proposed method improves performance on downstream tasks seems a bit overstretched.
Even when choosing the best settings across multiple hyperparameters, the performance improvement is modest (as seen in Table 1). Moreover, performance can deteriorate compared to the baseline depending on hyperparameter choices (as shown in Figure 3). Furthermore, adopting the proposed method incurs an additional cost of hyperparameter selection. Therefore, for practitioners aiming to employ IsoScore* for their problems, it's hard to advocate for the use of the proposed method. Of course, submissions to ICLR shouldn't be evaluated solely on empirical performance. However, in this paper, improving performance on downstream tasks is one of the main contributions; thus the lack of compelling experimental results will inevitably impact the paper's peer-review evaluation.

**Questions:**

- If anisotropy is a natural consequence of SDG, what is the significance of deliberately adding anisotropy as a regularization term? The computational cost increases slightly, and the disadvantage of increased hyperparameter tuning cost needs to be offset by some substantial benefits. If there are such advantages, the appeal of the proposed method would be enhanced.
- What is the significance of adjusting the hyperparameter $\zeta$ for shrinkage estimation when using IsoScore* as a regularization term? Even if the value of IsoScore is estimated to be on the lower side (Figure 2), if the estimated value has a monotonic relationship with the true value, wouldn't there be no issue in regularization? If the goal is to ensure differentiability, couldn't we just fix it at an appropriate value?

---

> ### Author Response · Authors · 2023-11-21
>
> $\textbf{Response to weaknesses}$
>
> $\textbf{1. The reasons for contradictions with prior research are unclear, weakening the persuasiveness of the main claim.}$
>
> We agree that more work needs to be done to determine the relationship between isotropy and word similarity tasks. However, most word similarity benchmarks simply calculate the cosine similarity of word embeddings and do not involve any model fine-tuning. Since I-STAR is a regularization algorithm that adjusts isotropy during fine-tuning, I-STAR may not be the most appropriate method to analyze the relationship between word similarity and isotropy. However, a future direction of work could continue the pre-training task of masked language modeling with an added I-STAR penalty to adjust for isotropy.
>
> Instead, our paper focuses on the relationship between isotropy and fine-tuning performance, as I-STAR is a regularization algorithm requiring fine-tuning tasks. However, our paper evaluates the relationship between isotropy and a model’s ability to rate the similarity of two sentences using the Semantic Textual Similarity Benchmark (STS-B). STS-B consists of sentence pairs annotated with a similarity score from 1-5.
>
> Although a majority of previous works have argued that isotropy is beneficial for LLM performance, there are several recent studies that support the claim that adjusting isotropy in fine-tuning hurts downstream model performance. Namely, Rajaee & Pilehvar 2021 show that adjusting for isotropy by removing principal components hurts model performance. Further, recent work by Rudman et al. 2023 demonstrates that a dimension’s ability to store task-specific knowledge strongly correlates with high variance in a given dimension.
>
> $\textbf{2. The claim that the proposed method improves performance on downstream tasks seems a bit overstretched.}$
>
> We agree that the performance increase of fine-tuning with I-STAR is modest. However, the primary aims of our paper are 1) to create a method for measuring isotropy that is differentiable and stable for mini-batch computations and 2) to analyze the relationship between isotropy and fine-tuning performance. In order to clarify the relationship between isotropy and performance, we have performed Pearson correlation analysis on the results of Figure 3. Initial analysis of our Pearson correlation analysis indicates that isotropy is indeed negatively correlated with performance in all cases (except ALBERT on COLA). Note that the small sample size of each entry (num_tuning_params x num_random_seeds = 35) may limit the generalizability of these results and cause some of the p values to be artificially high.
>
>     Model & QQP & RTE & MRPC & COLA & STS-B
>
>     BERT & $-0.573 / 0.001$ & $-0.418 / 0.012$ & $-0.115 / 0.511$ & $-0.227 / 0.190$ & $-0.677 / 0.001$
>
>     ALBERT & $-0.469 / 0.009$ & $-0.424 / 0.011$ & $-0.184 / 0.330$ & $0.195 / 0.261$ & $-0.624 / 0.004$
>
>     DistBERT & $-0.319 / 0.062$ & $-0.840 / 0.003$ & $-0.532 / 0.003$ & $-0.641 / 0.003$ & $-0.713 / 0.005$
>
> $\textbf{Response to questions}$
>
> $\textbf{1. If anisotropy is a natural consequence of SDG, what is the significance of deliberately adding anisotropy as a regularization term? }$
>
> Thank you for a great question! Previous works have argued that anisotropy helps models better generalize to unseen examples and is important in fine-tuning (Zhou et al. 2018). Our work empirically supports this claim by demonstrating a relationship between isotropy and performance. We hypothesize that the significance of adding our anisotropy penalty term helps models learn fine-tuning tasks faster. In order to verify our hypothesis, we are actively running an experiment where we track how quickly models converge when training with no regularization and different \lambda values in I-STAR.
>
> $\textbf{2. What is the significance of adjusting the hyperparameter for shrinkage estimation when using IsoScore* as a regularization term?}$
>
> Figure 7 in Section C of the Appendix addresses the significance of adjusting the shrinkage parameter $\zeta$ in I-STAR. In all cases, the optimal shrinkage parameter is in the interval [0.2,0.8]. Figure 7 shows that when $\zeta=0$ and we use (a differentiable version of) IsoScore as a regularization penalty model, performance decreases. Further, if we set $\zeta=0$, the isotropy penalty is computed directly from the shrinkage matrix. Namely, no mini-batch information is included in the loss function. When $\zeta=1$, performance is slightly lower than optimal $\zeta$, but significantly higher than when $\zeta=0$.  Results from this experiment demonstrate the need for IsoScore* in order to perform isotropic regularization.

---

### Official Review · Reviewer_ZAyV · 2023-11-02

**Soundness:** 3 good
**Presentation:** 4 excellent
**Contribution:** 3 good
**Rating:** 8
**Confidence:** 3

**Summary:**

This submission proposes a new method for measuring isotropy in neural models, improving over previous proposals and leading to a new regularization technique I-STAR. They show that LLMs actually seem to benefit from less isotropic internal representations, contrary to previous claims in the NLP literature.

**Strengths:**

The paper is well written and clear. The proposed improvement of IsoScore into IsoScore* is fairly straightforward. The fact that it is a more accurate and more convenient estimate of isotropy in LLMs is argued very well and supported by some empirical results.

The paper convincingly argues that isotropy and its impact on performance are not properly understood in NLP, which is a very significant contribution.

Experimental results mostly support the arguments in the paper (more comments below).

**Weaknesses:**

* There is no significance testing on the results (Table 1) but there are error bars (good!) — these seem to indicate that most differences outlined are hardly significant (e.g. RTE, 72.56+/-1.29 vs 71.34+/-0.91). This makes it difficult to get a clear picture of the resulting effect of decreasing isotropy.
* Similarly Fig. 3 is difficult to interpret — there are clear decreasing trends in some plots, not so much in most of them.

**Questions:**

* p.4: Presumably all norms in steps 6 and 7 of Algorithm 1 are 2-norms?
* p.4: Is RDA "regularized discriminant analysis"? It is only introduced on p.5.
* p.4: As \Sigma_S is computed from a large number of tokens and re-estimated after each epoch, this is presumably quite costly computationally. Could you comment on that? As you are computing these anyway, why not use that directly to estimate isotropy? This would essentially correspond to \zeta=1, which is not tested here as far as I can tell.
* p.6: "Section F" -- do you mean Section D?
* p.7: I do not clearly see Fig. 4 as supporting the claim that CosReg only alters the mean of activations (all means seem to be ~zero on the plots, with one outlier). Maybe a plot of the distribution would support this claim better?
* p.9: Why are -1 and +1 lambdas excluded from Fig. 5? -1 seems the most popular choice in Fig. 2.
* Regularization to a manifold in parameter space is well studied in Machine Learning, and indeed supports the argument that anisotropy may benefit performance. This is also linked to the idea that there is the level of representativity in the model that must match the intrinsic dimension or complexity in the data. Fig. 5 actually seems to show roughly consistent intrinsic dimensions for the three models used here. Is there a way you could put this in perspective with e.g. model size? Showing for exemple how anisotropy favors a roughly stable internal dimensionality in parameter space as model size increases?

---

> ### Author Response · Authors · 2023-11-21
>
> $\textbf{Response to weaknesses}$
>
> $\textbf{1.}$ We will run significance testing for each of the model/task setups and display the significance testing results in the paper's main body.
>
> $\textbf{2.}$ In order to clarify the relationship between isotropy and performance, we have performed Pearson correlation analysis on the results of Figure 3. Initial analysis of our Pearson correlation analysis indicates that isotropy is indeed negatively correlated with performance in all cases (except ALBERT on COLA). Note that the small sample size of each entry (num_tuning_params x num_random_seeds = 35) may limit the generalizability of these results and cause some of the p values to be artificially high.
>     Model & QQP & RTE & MRPC & COLA & STS-B
>
>     BERT & $-0.573 / 0.001$ & $-0.418 / 0.012$ & $-0.115 / 0.511$ & $-0.227 / 0.190$ & $-0.677 / 0.001$
>
>     ALBERT & $-0.469 / 0.009$ & $-0.424 / 0.011$ & $-0.184 / 0.330$ & $0.195 / 0.261$ & $-0.624 / 0.004$
>
>     DistBERT & $-0.319 / 0.062$ & $-0.840 / 0.003$ & $-0.532 / 0.003$ & $-0.641 / 0.003$ & $-0.713 / 0.005$
>
> $\textbf{Response to questions:}$
>
> $\textbf{1. Pseudo-Code Clarification.}$
>
> Steps 6 &  7 of Algorithm 1 are L2 norms. We will add this to the pseudo-code of the algorithm.
>
> $\textbf{2. RDA Clarification.}$
>
> RDA is indeed a Regularized Discriminant Analysis, which we will introduce earlier in the revised version of our paper.
>
> $\textbf{3.}$  $\Sigma\_{s}$ $\textbf{computation cost and impact of}$ $\zeta$
>
> Since we only need to collect a sample of 250k tokens, the partial forward pass on the dataset. Computing the isotropy penalty only on \Sigma_S is equivalent to setting $\zeta=1$. The downside to setting $\zeta=1$ is that no isotropy information about the mini-batches of data will be included in the backward pass. Section C of the Appendix shows the results of fine-tuning BERT, ALBERT, and DistilBERT on RTE using different values of \zeta. Figure 7 (Sec C) demonstrates that setting $\zeta=1$ results in a marginal decline in performance compared to incorporating mini-batch isotropy information during gradient descent.
>
> $\textbf{ 4. I do not clearly see Fig. 4 as supporting the claim that CosReg only alters the mean of activations.}$
>
> Table 2 in Section F of the Appendix lists the optimal I-STAR hyperparameter values for all tasks and models.
> Apologies for the lack of clarity. Figure 4 shows the mean value for each dimension in the model. While most of the dimensions are centered around zero, a few dimensions have mean values very far away from the origin. Figure 4 demonstrates that CosReg simply alters the mean of model representations in a single basis dimension instead of altering the isotropy of representations. In each “base” case where no CosReg is performed (shown in green), one basis dimension has mean values far from the origin. Fine-tuning with tuning parameters with CosReg setting $\lambda=1$ has the effect of moving the mean value in that dimension to zero while using CosReg with $\lambda=-1$ pushes that outlier basis dimension even further away from the origin. For example, when fine-tuning with ALBERT on QNLI, the basis dimension with the largest mean changes from an average value of 11.72 (base) to 27.33 when using CosReg -1 and decreases to 2.84 when using CosReg +1. Altering the mean in this single direction with CosReg changes the cosine similarities from 0.003 ($\lambda=-1$) to 0.998 ($\lambda=1$)
>
> $\textbf{5. Adding}$ $\pm1 \  \lambda$
>
> We have updated Figure 5 to add \lambda values of +1 and -1.
>
> $\textbf{6. Does anisotropy favor a roughly stable internal dimensionality in parameter space as model size increases?}$
>
> That is a very interesting question! Based on your comment, there are two potential experiments to design. First, we can fix the task and fine-tune different model sizes for that task. For example, we can fine-tune BERT-base & BERT-large as well as all of the available ALBERT model sizes from ALBERT-base (12m parameters) to ALBERT-xxl (235M parameters) with different tuning parameter values in I-STAR and calculate the intrinsic dimension of the fine-tuned models. However, given that BERT, DistilBERT, and ALBERT all have a different number of parameters, an interesting follow-up experiment would be to run the experiments in Figure 5 for all of the tasks considered in this paper to verify if there is a desirable intrinsic dimension for model representations for each of the different datasets.

---

> > ### Comment · Reviewer_ZAyV · 2023-11-21
> > **Thanks for the replies**
> >
> > Thank you for the clarifications and additional information. Significance tests may help, but the significance of the performance improvements (also noted by reviewer tWNn) is still mildly convincing.
> > The clarification re. Fig. 4 does not really address the issue, however. What I *see* in Fig. 4 is that, apart from one outlier, the plotted values seem roughly unchanged for BERT and DistilBERT. Maybe a plot *excluding* the outlier would help reinforce the point which I think is that the green points should lie roughly between the yellow and the blue. Which I think is a fairly obvious consequence of the regularizer, but is not really apparent in all plots currently.
> >
> > After reading the other reviews and replies, I am still leaning positive on this paper. I agree with tWNn that there are legitimate concerns with how persuasive the main claim is (that anisotropy is preferable). However, I find the paper convincing in its claim that isotropy is not clearly and universally desirable. If it is not settling the debate in a decisive manner, it has the merit of pushing convincing arguments and some evidence that clearly question current understanding.

---

### Official Review · Reviewer_GH3h · 2023-11-08

**Soundness:** 2 fair
**Presentation:** 2 fair
**Contribution:** 3 good
**Rating:** 8
**Confidence:** 2

**Summary:**

This study investigates the connection between isotropy and model performance by employing 3 distinct LLMs and 9 fine-tuning tasks. It introduces I-STAR, a method for adjusting model isotropy based on a novel differentiable metric called IsoScore*.

Surprisingly, the study's findings contradict the prevailing notion in NLP literature. That is, it demonstrates experimentally that discouraging isotropy leads to better performance across the different models and downstream tasks.

**Strengths:**

This paper challenges the dominant belief in NLP literature showing that anisotropy is beneficial. Its findings have the potential to significantly influence future research directions in the field.

It also introduces new way to compute the Isotropy in models. The authors conducted a set of experiments to show its efficiency comparing it to CosReg.

**Weaknesses:**

If we see a trend in Figure 3 on how higher IsoScore* leads to lower accuracy, some correlation and significance score should be added to support this claim.

**Questions:**

Could you add in the Appendix the same experiments done in Figure 3 but for CosReg and also the same in Figure 4 but for IsoScore*? This will give a better intuition of how these 2 different methods work.

---

> ### Author Response · Authors · 2023-11-21
>
> 1. In order to clarify the relationship between isotropy and performance, we have performed Pearson correlation analysis on the results of Figure 3. Initial analysis of our Pearson correlation analysis indicates that isotropy is indeed negatively correlated with performance in all cases (except ALBERT on COLA). Note that the small sample size of each entry (num_tuning_params x num_random_seeds = 35) may limit the generalizability of these results and cause some of the p-values to be artificially high.
>
>     Model & QQP & RTE & MRPC & COLA & STS-B
>
>     BERT & $-0.573 / 0.001$ & $-0.418 / 0.012$ & $-0.115 / 0.511$ & $-0.227 / 0.190$ & $-0.677 / 0.001$
>
>     ALBERT & $-0.469 / 0.009$ & $-0.424 / 0.011$ & $-0.184 / 0.330$ & $0.195 / 0.261$ & $-0.624 / 0.004$
>
>     DistBERT & $-0.319 / 0.062$ & $-0.840 / 0.003$ & $-0.532 / 0.003$ & $-0.641 / 0.003$ & $-0.713 / 0.005$
>
>
> 2. The updated version of our paper will contain the results of Figure 3 for CosReg and Figure 4 for IsoScore*.

---

### Meta-Review · Area_Chair_mx9L · 2023-12-06

**Metareview:**

This paper proposes a regularization technique called I-STAR that can increase or decrease levels of isotropy in embedding space during finetuning. I-STAR is based on a newly introduced measure of isotropy called IsoScore* (based on IsoScore). In contrast to previous works, the paper has found that decreasing isotropy in contextualized embeddings actually improves performance for the tasks/models evaluated.

All reviewers found the (unexpected) finding of the paper (showing anisotropy is beneficial) intriguing. While reviewers GH3h and ZAyV are convinced by the experimental results and only raise minor concerns, reviewer tWNn are less convinced by some of the claims. In particular, the proposed work evaluates on the GLUE benchmarks instead of word similarity tasks, making it less comparable with previous works, therefore weakening its finding. It might be helpful for the authors to further clarify such differences.

**Justification For Why Not Higher Score:**

Some reviewers are not fully convinced by the findings.

**Justification For Why Not Lower Score:**

The unexpected finding that anisotropy is beneficial for finetuning is interesting and could be good contribution to the research community by itself.

---

### Decision · Program_Chairs · 2024-01-16

Accept (poster)